# Suicidal ideation following self-reported COVID-19-like symptoms or serology-confirmed SARS-CoV-2 infection in France: A propensity score weighted analysis from a cohort study

**Camille Davisse-Paturet**[1]*, **Massimiliano Orri**[2], **Stéphane Legleye**[1,3], **Aline-Marie Florence**[4], **Jean-Baptiste Hazo**[5], **Josiane Warszawski**[6], **Bruno Falissard**[1], **Marie-Claude Geoffroy**[2,7], **Maria Melchior**[4], **Alexandra Rouquette**[1,6], **the EPICOV study group**¶

1 Université Paris-Saclay, Inserm, UVSQ, CESP, Paris, France, 2 McGill University, Department of Psychiatry, Montreal, Québec, Canada, 3 Ensai, Bruz, France, 4 Sorbonne University, Inserm, Pierre Louis institute of Epidemiology and Public Health, Paris, France, 5 French Ministry of Solidarity and Health, Drees, Paris, France, 6 APHP, Paris-Saclay University, Department of Epidemiology and Public Health, Le Kremlin-Bicêtre, France, 7 McGill University, Department of Educational and Counselling Psychology, Québec, Canada

¶ Membership of EpiCoV study group is provided in the Supporting information file S1 Acknowledgements.
* camille.davisse-paturet@inserm.fr

**Academic Editor:** Toshiaki A. Furukawa, Kyoto University Graduate School of Medicine / School of Public Health (current) and Nagoya City University Graduate School of Medical Sciences (at the time of the study), JAPAN

## Abstract

### Background

A higher risk of suicidal ideation associated with self-report of Coronavirus Disease 2019 (COVID-19)-like symptoms or COVID-19 infection has been observed in cross-sectional studies, but evidence from longitudinal studies remains limited. The aims of this study were 2-fold: (1) to explore if self-reported COVID-19-like symptoms in 2020 were associated with suicidal ideation in 2021; (2) to explore if the association also existed when using a biological marker of Severe Acute Respiratory Syndrome Coronavirus 2 (SARS-CoV-2) infection in 2020.

### Methods and findings

A total of 52,050 participants from the French EpiCov cohort were included (median follow-up time = 13.7 months). In terms of demographics, 53.84% were women, 60.92% were over 45 years old, 82.01% were born in mainland France from parents born in mainland France, and 59.38% completed high school. COVID-19-like symptoms were defined as participant report of a sudden loss of taste/smell or fever alongside cough, shortness of breath, or chest oppression, between February and November 2020. Symptoms were self-reported at baseline in May 2020 and at the first follow-up in Autumn 2020. Serology-confirmed SARS-CoV-2 infection in 2020 was derived from Spike protein ELISA test screening in dried-blood-spot samples. Samples were collected from October 2020 to March 2021, with 94.4% collected

**Data Availability Statement:** All anonymous aggregated data regarding the results presented in this paper are available online and on supporting information files. The non-aggregated individual data cannot be shared publicly because of European Regulation 2016/679. Nonetheless, these data can be made available after submission to approval of French Ethics and Regulatory Committee procedure (Comité du Secret Statistique, CESREES and CNIL). The access procedure is available on the Centre of Secured Access to Data website (https://www.casd.eu/).

**Funding:** The present work was supported by a research grant from the department of research, studies, evaluation and statistics (Direction de la Recherche, des Etudes, de l'Evaluation et des Statistiques, Drees) of the French Ministry for Research attributed to AR (grant number R21094LL). The EpiCov study received institutional fundings from Inserm (Institut National de la Santé et de la Recherche Médicale), the French Ministry for Research and its department of research, studies, evaluation and statistics (Direction de la Recherche, des Etudes, de l'Evaluation et des Statistiques, Drees), the French Ministry for Health, and the Région Ile de France. https://drees.solidarites-sante.gouv.fr/article/observatoire-national-du-suicide https://www.inserm.fr/ https://www.enseignementsup-recherche.gouv.fr/fr https://drees.solidarites-sante.gouv.fr/ https://solidarites-sante.gouv.fr/ https://www.iledefrance.fr/ The funders had no role in study design, data collection and analysis, decision to publish, or preparation of the manuscript.

**Competing interests:** The authors have declared that no competing interests exist.

**Abbreviations:** ATT, average treatment effect on the treated; BMI, body mass index; CATI, computer-assisted telephone interview; CAWI, computer-assisted web interview; COVID-19, Coronavirus Disease 2019; DAG, directed acyclic graph; INSEE, The National Institute for Statistics and Economic Studies; IPW, inverse probability weight; PCR, polymerase chain reaction; RR, relative risk; SARS-CoV-2, Severe Acute Respiratory Syndrome Coronavirus 2; SMD, standardized mean difference.

in 2020. Suicidal ideation since December 2020 was self-reported at the second follow-up in Summer 2021. Associations of self-reported COVID-19-like symptoms and serology-confirmed SARS-CoV-2 infection in 2020 with suicidal ideation in 2021 were ascertained using modified Poisson regression models, weighted by inverse probability weights computed from propensity scores. Among the 52,050 participants, 1.68% [1.54% to 1.82%] reported suicidal ideation in 2021, 9.57% [9.24% to 9.90%] had a serology-confirmed SARS-CoV-2 infection in 2020, and 13.23% [12.86% to 13.61%] reported COVID-19-like symptoms in 2020. Self-reported COVID-19-like symptoms in 2020 were associated with higher risks of later suicidal ideation in 2021 (Relative Risk$_{ipw}$ [95% CI] = 1.43 [1.20 to 1.69]), while serology-confirmed SARS-CoV-2 infection in 2020 was not (RR$_{ipw}$ = 0.89 [0.70 to 1.13]). Limitations of this study include the use of a single question to assess suicidal ideation, the use of self-reported history of mental health disorders, and limited generalizability due to attrition bias.

## Conclusions

Self-reported COVID-19-like symptoms in 2020, but not serology-confirmed SARS-CoV-2 infection in 2020, were associated with a higher risk of subsequent suicidal ideation in 2021. The exact role of SARS-CoV-2 infection with respect to suicide risk has yet to be clarified. Including mental health resources in COVID-19-related settings could encourage symptomatic individuals to care for their mental health and limit suicidal ideation to emerge or worsen.

## Author summary

### Why was this study done?

- There is a need to investigate suicide-related outcomes in individuals exposed to virus responsible for epidemics.
- Coronavirus Disease 2019 (COVID-19) infection seems associated with a higher risk of suicidal ideation, but studies exploring the association over time are limited.

### What did the researchers do and find?

- We explored if COVID-19-like symptoms in 2020, as reported by participants, and Severe Acute Respiratory Syndrome Coronavirus 2 (SARS-CoV-2) infection in 2020, as confirmed by serological tests, were associated with a higher risk of suicidal ideation in 2021 in 52,050 participants from a French longitudinal study from the general population, followed three times from May 2020 to July 2021 in France.
- Among these participants, reporting COVID-19-like symptoms in 2020 was associated with a higher risk of reporting suicidal ideation in 2021 (relative risk [95% confidence interval] 1.43 [1.20 to 1.69]), while having a serologically confirmed SARS-CoV-2 infection in 2020 was not associated with a higher risk of reporting suicidal ideation in 2021 (0.89 [0.70 to 1.13]). These results account for sociodemographic and health-related factors.

**What do these findings mean?**

- Individuals experiencing COVID-19-like symptoms in the first year of the pandemic were at higher risk of later suicidal ideation, but this association was not observed for serologically confirmed SARS-CoV-2 infection. Thus, further study is needed to confirm the role of the virus in relation to suicide risk.

- From a public health perspective, short communication on what to do when someone is experiencing poor mental health, whether directly after the onset of symptoms or a few months later, may be beneficial to address increase in suicidal ideation.

## Introduction

Since the beginning of the Coronavirus Disease 2019 (COVID-19) pandemic, mental health specialists have raised concerns regarding the possibility of an increase in the risk of suicidal behaviors among persons recovering from COVID-19 [1–3]. Moreover, as shown by a recent systematic review, exploration of suicide-related outcomes in individuals infected by infectious threat responsible for epidemics are needed [4].

Regarding potential biological pathways, the exact role of Severe Acute Respiratory Syndrome Coronavirus 2 (SARS-CoV-2) with regard to mechanisms involved in suicide risk has yet to be demonstrated. Nonetheless, the virus's ability to invade the central nervous system through fixation on angiotensin-converting enzyme 2 receptors [5], or to inflict brain damage through hyperinflammation [6], are potential candidates.

From a public health point of view, suicide risk related to COVID-19 disease was first supported by findings from cross-sectional studies and case series [7,8], and evidence from longitudinal studies are now emerging. A systematic review ascertaining the risk of suicidal and self-harm thoughts and behaviors among persons recovering from SARS-CoV-2 infection identified 11 relevant studies conducted between January 2020 and July 2021 and representing eight separate samples [9]. Eight out of the 11 studies reported elevated risk of suicidal or self-harm thoughts after SARS-CoV-2 infection. Unfortunately, these studies were quite heterogenous in terms of suicide risk assessment, study population, or design. Of note, the only study with a longitudinal design found that self-reported suspicion or diagnosis of COVID-19 was associated with an elevated risk of self-harm thoughts or behaviors over 59 weeks since the end of March 2020 [10]. Additionally, a longitudinal study from Australia also reported an increased risk of suicidal ideation in the three months following exposure to COVID-19, even when controlling for the pandemic's impact on employment and financial distress [11]. Yet, as explained by the authors, the situation in Australia at the time of assessment, between March and June 2020, was quite different from that in other parts of the world, including Europe. In a paper gathering data from seven longitudinal cohorts from six Northern-Europe countries, the number of days spent in bed due to SARS-CoV-2 infection was cross-sectionally associated with mental health outcomes in a dose-effect fashion [12]. Participants not bedridden by the infection were at lower risk of both depressive and anxiety symptoms than those who were not infected, while those bedridden for seven days or more were at higher risk. As depressive and anxiety symptoms are known trigger of suicidal behaviors [13], taking into account both the infection status and its associated symptoms might therefore be relevant when studying suicide risk. Moreover, as both COVID-19 disease and suicidal behaviors are related to individuals'

sociodemographic and health characteristics [13,14], comprehensive information regarding theses aspects are needed to address the association between COVID-19 disease and suicidal behavior. As causal inference is not possible in observational studies, the use of propensity scores weighting methods, which balance all selected covariates between exposure groups, can be a mean of getting closer to assessment of direct association between an exposure and an outcome [15].

Our aim was therefore to study the association of self-reported COVID-19-like symptoms in 2020 with suicidal ideation in 2021. We also aimed to explore if the association could be found when using serology-confirmed SARS-CoV-2 infection. We used data from a French cohort from the general population and accounted for a wide range of sociodemographic and health-related factors through inverse probability weighting.

## Methods

### Study population

The Epidémiologie et conditions de vie sous le COVID-19 (EpiCov) study is a longitudinal, French cohort from the general population, aiming to provide information on the virus' dissemination and the pandemic's consequences on the daily life and health of individuals [16]. Eligibility criteria were to be at least 15 years of age in 2020, to reside in mainland France or three oversea territories (Martinique, Guadeloupe, and Réunion), and to not live in a medical retirement home or a jail. A total of 371,000 individuals were randomly selected from France's national tax database, with an expected participation rate of about 50% and a sampling design overrepresenting less densely populated and more socioeconomically disadvantaged areas [17].

Data collection methods included both self-computer-assisted web interviews (CAWIs) and computer-assisted telephone interviews (CATIs). From the 371,000 randomly selected individuals invited to participate, 36.22% (134,391) actually participated at baseline in May 2020 (02/05/2020 to 02/06/2020). The first follow-up took place in Autumn 2020 (26/10/2020 to 14/12/2020, 107,759 participants), and the second follow-up took place in Summer 2021 (24/06/2021 to 09/08/2021, 85,074 participants). The EpiCov study timeline as well as data collected and used are resumed in Fig 1.

The EpiCov study received approval from an ethics committee (Comité de Protection des Personnes Sud Méditerranée III 2020-A01191-38) and from France's National Data Protection Agency (Commission Nationale Informatique et Libertés, CNIL, MLD/MFI/AR205138).

The present study does not have a registered prospective protocol. An unpublished, informal analysis plan was made and discussed among study authors prior to the implementation of statistical analyses.

### Outcome: Suicidal ideation in 2021

At the second follow-up in Summer 2021, suicidal ideation were ascertained with the question, "Since December 2020, have you thought about 'killing yourself by suicide'?" Killing oneself by suicide is the closest translation of the actual French word used in the questionnaire "se suicider." The outcome of interest was a binary variable representing the occurrence of suicidal ideation at least once between December 2020 and July 2021 (yes versus no).

### Exposures

**Serology-confirmed SARS-CoV-2 infection in 2020.**   The exact methodology for serology testing has been described elsewhere [16]. Briefly, consent to participate to SARS-CoV-2

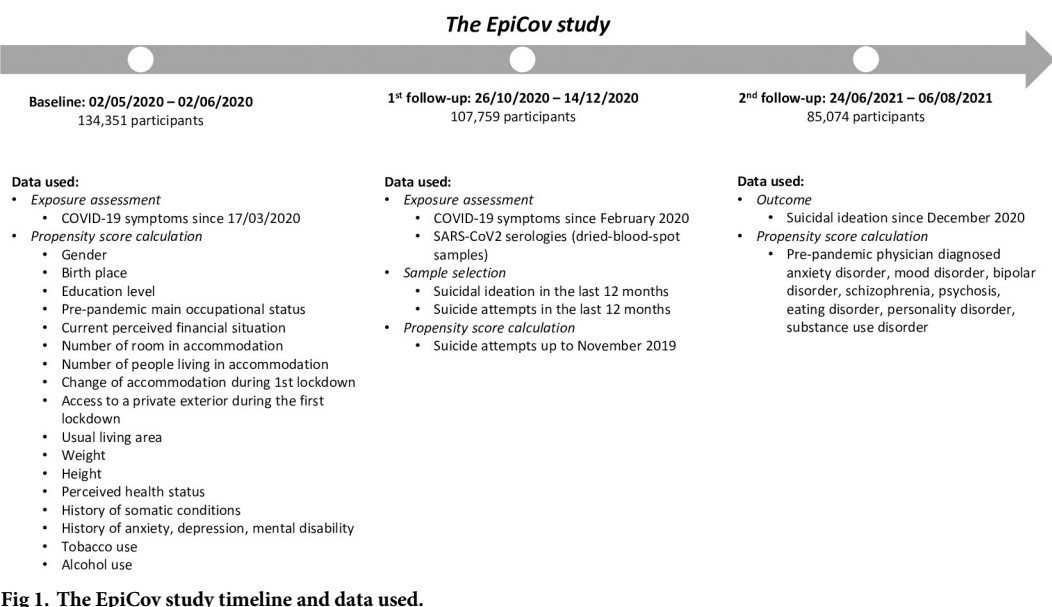

**Fig 1. The EpiCov study timeline and data used.**

serology testing with blood sampling kit was collected at the first follow-up in Autumn 2020. A total of 63,524 dried-blood-spot samples were screened for antibodies against SARS-CoV-2's spike protein S1 domain, with a commercial ELISA kit. Samples with a serology-confirmed SARS-CoV-2 infection had an optical density ratio of at least 0.7. We pooled suspicious serologies (i.e., with an optical density ratio between 0.7 and 1.1) with positive ones (i.e., optical density ratio >1.1) as a decline in circulating antibodies might occur with time [18]. Of note, 94.4% of blood samples were collected before January 2021 and the start of the vaccination campaign in France. Serology-confirmed SARS-CoV-2 infections were therefore unlikely to be due to vaccination.

**Self-reported COVID-19-like symptoms in 2020.** In this study, COVID-19-like symptoms were symptoms described as most suspicious in 2020 by the French Public Health Agency. They referred to any unusual episode of sudden loss of taste/smell or any unusual episode of fever alongside a cough, shortness of breath, or chest oppression. At baseline in May 2020, participants reported COVID-19-like symptoms since the 17th of March 2020. At first follow-up in Autumn 2020, they reported COVID-19-like symptoms since February 2020. Pooling the two, this study's self-reported COVID-19-like symptoms (yes versus no) were self-report of COVID-19-like symptoms at least once from February 2020 to Autumn 2020.

## Propensity score covariates

Covariate selection for propensity score modeling was based on current literature, including recommendations from the International COVID-19 Suicide Prevention Research Collaboration [19]. In accordance with propensity score methodology [20], included factors were related to both COVID-19 and suicidal ideation or only to suicidal ideation [8]. Directed acyclic graphs (DAGs) supported framework conceptualization for the assessment of suicidal ideation related to COVID-19 and minimize bias though appropriate covariate selection [21,22] (S3 Supporting Information).

**Sociodemographic and health covariates.** The following sociodemographic covariates were ascertained at baseline and included in propensity scores: gender (man, woman), age (years), participant's and participant's parents' place of birth (participant and parents born in

mainland France, participant or parents born in oversea territories, participant born in France of parents born abroad, participant born abroad), highest educational attainment (none, lower secondary school certificate, professional certificate, higher secondary school certificate, bachelor degree or equivalent, Master degree or more), occupational grade (employed, student, unemployed, retired, other including housemakers), perceived financial situation (comfortable, decent, short, difficult or unbearable without taking loans), physical space in participant's usual accommodation (less than one room per person, yes or no), residence not in usual housing during the first lockdown (yes or no, the first lockdown lasted from the 17/03/2020 to the 11/05/2020), access to safe outdoor space during the first lockdown (balcony or garden, including common ones, yes or no), and usual living area, ranked according to the intensity of the first COVID-19 wave in France (less affected areas, Grand-Est, Hauts-de-France, Ile-de-France). More information regarding the first COVID-19 epidemic wave in France are available in S2 Supporting Information.

The following health-related covariates were also ascertained: perceived general health status at baseline (very good to good, quite good, poor to very poor), baseline body mass index (BMI; less than 18.5 kg/m$^2$, between 18.5 and less than 25 kg/m$^2$, between 25 and less than 30 kg/m$^2$, 30 kg/m$^2$ or more), pre-pandemic somatic conditions (yes or no), pre-pandemic mental health disorder (yes or no), baseline tobacco use (current, past, never), and baseline alcohol use (daily, often, occasional, rare, never). Pre-pandemic mental health disorder included self-reported anxiety, depression, and mental disability, assessed at baseline, history of at least one suicide attempt before November 2019, assessed at first follow-up, and self-report of at least one physician diagnosis of anxiety, mood, bipolar, eating, personality or substance use disorder, or schizophrenia before the pandemic, assessed at second follow-up. A detailed description of the pre-pandemic mental health disorder covariate is available in S1 Supporting Information.

**Available indicators.**   The following indicators, made available by The National Institute for Statistics and Economic Studies (INSEE), were taken into account: deciles of household income per consumption unit studied as a five-category covariate (less resourceful, medium-low, medium, medium-high, wealthiest), household structure (single, couple without children, couple with children, single-parent, participant living with parents, complex household), urban density of living area (oversea territories, less than 2,000 urban units, between 2,000 and 1,999,999 urban units, Paris area), residence in a deprived neighborhood (yes or no), and hospitalization rates in place of residence during the first lockdown (lowest, medium-low, medium-high, highest quartile). An urban unit is a built area with less than 200 meters between two buildings, comprising at least 2,000 inhabitants. A deprived neighborhood is an administrative category, describing an area where particular budgetary efforts are made by the State to tackle inequalities regarding education, early life care, housing and living conditions, employment, social cohesion, security, and crime prevention.

## Statistical analyses

Suicidal ideation in 2020 as well as life course suicide attempts were assessed at the first follow-up in Autumn 2020 (Fig 1). The related questions were, respectively, "In the last 12 months, have you thought about 'killing yourself by suicide'?" and "In your lifetime, have you ever attempted suicide? If so, when was the last one?" Yet, mediators are not supposed to be included in propensity score calculation. Suicidal ideation and suicide attempts in 2020 could act as mediator in the association of self-reported COVID-19-like symptoms in 2020 and serology-confirmed SARS-CoV-2 infection in 2020 with suicidal ideation in 2021. But, in EpiCov, no data were available regarding the first occurrence of COVID-19-like symptoms in 2020 or

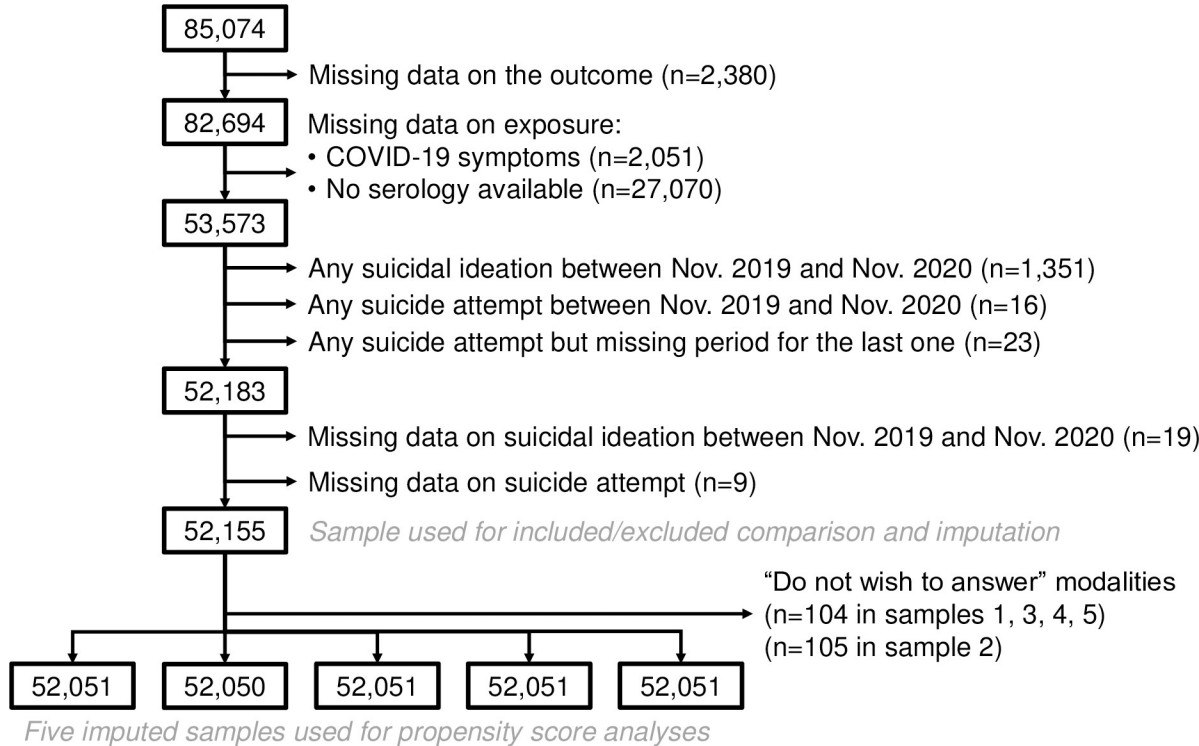

**Fig 2. Flow chart.**

date of SARS-CoV-2 infection in 2020. We could therefore not ascertain if suicidal ideation or suicide attempt in 2020 occurred before or after the two COVID-19 exposures. To ensure no mediator was included in propensity score calculation, participants reporting suicidal ideation or a history of suicide attempt in 2020 or who did not provide information on the timing of their last suicide attempt had to be removed from the statistical analyses. Participants who reported a last suicide attempt before 2020 were included in the pre-pandemic mental health disorder covariate. Excluded participants are detailed in Fig 2. Do not wish to answer modalities had to be removed because of small sample size.

**Descriptive statistics.** Study weights were applied to all descriptive statistics (Tables 1, 2, 3 and S1) in order to take EpiCov's design and attrition bias into account. Briefly, the study weights accounted for demographic and socioeconomic indicators potentially linked to response probability and made available by INSEE from the tax data base. Study weights were also calibrated on margins of the general population according to census data and population projections [23]. The numerators and denominators for each descriptive table are available in the supporting tables file (S2, S3, S4 and S5 Tables). Because descriptive statistics were weighted by study weights the presented percentages do not reflect the division of numerators by denominators. Except for included/excluded comparison, all presented percentages were pooled from five imputed data sets (more details below, SAS MIANALYZE procedure).

**Inverse probability weighting and modified Poisson regression models.** First, propensity scores associated with (a) reporting COVID-19-like symptoms or (b) having a serology-confirmed SARS-CoV-2 infection were computed using logistic regression models based on the covariates described above. Then, propensity scores were included in statistical models using inverse probability weights (IPWs; S5 Supporting Information). Balance after IPWeighting was considered satisfactory if (1) absolute standardized mean differences (SMDs) between

**Table 1. Prevalence of exposures and covariates according to suicidal ideation in 2021, weighted by study weights, pooled from five imputed data sets.**

| | Suicidal ideation in 2021 | |
| --- | --- | --- |
| | No (% [95% CI]) | Yes (% [95% CI]) |
| *Serology-confirmed SARS-CoV-2 infection in 2020* | | |
| No | 90.42% [90.09–90.76] | 90.65% [88.21–93.08] |
| Yes | 9.58% [9.24–9.91] | 9.35% [6.92–11.79] |
| *Self-reported COVID-19 symptoms in 2020* | | |
| No | 86.93% [86.55–87.31] | 77.39% [74–80.78] |
| Yes | 13.07% [12.69–13.45] | 22.61% [19.22–26] |
| *Gender* | | |
| Men | 46.17% [45.59–46.75] | 45.71% [41.48–49.94] |
| Women | 53.83% [53.25–54.41] | 54.29% [50.06–58.52] |
| *Age (years)* | | |
| 15–25 | 11.03% [10.67–11.38] | 24.42% [20.68–28.16] |
| 26–45 | 27.72% [27.22–28.23] | 34.1% [30.14–38.07] |
| 46–65 | 35.02% [34.49–35.54] | 29.09% [25.57–32.6] |
| >65 | 26.23% [25.66–26.8] | 12.39% [9.22–15.56] |
| *Place of birth* | | |
| Participant and parents born in mainland France | 82% [81.5–82.49] | 82.96% [79.67–86.25] |
| Participants or parents born in oversea territories | 1.96% [1.8–2.11] | 2.21% [0.6–3.82] |
| Participants born in France from parents born abroad | 8.6% [8.25–8.94] | 9.31% [7.03–11.6] |
| Participants born abroad | 7.45% [7.07–7.82] | 5.51% [3.4–7.63] |
| *Highest academic level* | | |
| No diploma | 8.26% [7.81–8.71] | 4.69% [2.54–6.84] |
| Certificate from junior high at most | 13.32% [12.84–13.8] | 12.35% [9.36–15.35] |
| Professional certificate | 19.21% [18.75–19.66] | 13.61% [10.64–16.57] |
| Baccalaureat | 19.49% [19.05–19.92] | 23.24% [19.54–26.93] |
| 2 to 4 years after bac | 25.3% [24.86–25.75] | 27.4% [23.93–30.87] |
| At least 5 years after bac | 14.42% [14.09–14.76] | 18.71% [15.64–21.78] |

*Pooled weighted % [95% confidence interval]*

*Individuals in each cell in each of the imputed data sets are available in* S3 Table

| | Suicidal ideation in 2021 | |
| --- | --- | --- |
| | No (% [95% CI]) | Yes (% [95% CI]) |
| *Employment status before first LD* | | |
| Employed | 47.99% [47.41–48.56] | 50.21% [46.02–54.4] |
| Students | 8.54% [8.23–8.84] | 21.51% [17.92–25.1] |
| Unemployed | 4.63% [4.36–4.91] | 4.46% [2.63–6.29] |
| Retired | 30.62% [30.05–31.19] | 15.14% [11.85–18.44] |
| Other situations | 8.23% [7.88–8.58] | 8.68% [6.51–10.84] |
| *Perceived financial situation* | | |
| Comfortable | 16.15% [15.78–16.52] | 15.85% [12.83–18.86] |
| Decent | 43.5% [42.93–44.06] | 36.41% [32.45–40.37] |
| Short | 30.99% [30.43–31.56] | 32.05% [28.14–35.97] |
| Hard or cannot make it without debt | 9.36% [8.96–9.76] | 15.69% [12.35–19.03] |
| *Household structure* | | |
| Single | 16.74% [16.24–17.24] | 15.8% [12.88–18.72] |
| Couple without children | 32.31% [31.77–32.84] | 21.32% [17.74–24.91] |
| Couple with children | 28.81% [28.31–29.3] | 25.35% [21.96–28.73] |

*(Continued)*

**Table 1.** (Continued)

|  | Suicidal ideation in 2021 | |
| --- | --- | --- |
|  | **No (% [95% CI])** | **Yes (% [95% CI])** |
| Single-parent family | 7.16% [6.85–7.47] | 9.86% [7.36–12.36] |
| Child living at family house | 7.78% [7.47–8.08] | 12.64% [10–15.28] |
| Complex household | 7.21% [6.91–7.52] | 15.04% [11.58–18.49] |
| *Household income per consumption units* | | |
| Less resourceful | 13.75% [13.29–14.22] | 18.17% [14.73–21.6] |
| Medium-low | 16.6% [16.08–17.11] | 18.19% [14.6–21.78] |
| Medium | 19.89% [19.41–20.37] | 18.74% [15–22.47] |
| Medium-high | 23.73% [23.26–24.21] | 21.89% [18.14–25.65] |
| Wealthiest | 26.03% [25.58–26.48] | 23.01% [19.59–26.44] |
| *Physical space in usual accommodation* | | |
| No | 93.37% [93.05–93.69] | 90.78% [88.1–93.46] |
| Yes | 6.63% [6.31–6.95] | 9.22% [6.54–11.9] |
| *Changed accommodation for first LD* | | |
| No | 95.11% [94.87–95.36] | 89.81% [87.22–92.4] |
| Yes | 4.89% [4.64–5.13] | 10.19% [7.6–12.78] |
| *Access to safe outdoor space* | | |
| No | 8.62% [8.26–8.98] | 11.42% [8.89–13.95] |
| Yes | 90.45% [90.07–90.83] | 87.75% [85.15–90.35] |
| Other situations | 0.93% [0.8–1.06] | 0.83% [0.15–1.51] |
| *Usual area of residence* | | |
| Less affected area | 65.73% [65.18–66.27] | 62.3% [58.2–66.39] |
| Grand-Est | 8.75% [8.44–9.06] | 5.96% [4.12–7.79] |
| Hauts-de-France | 8.31% [7.98–8.64] | 10.04% [7.53–12.55] |
| Ile-de-France | 17.21% [16.78–17.65] | 21.71% [18.11–25.3] |
| *Urban density (urban units)* | | |
| oversea territories | 1.27% [1.16–1.38] | 1.07% [0.47–1.67] |
| Rural (less than 2,000) | 24.09% [23.6–24.58] | 22.74% [19.1–26.38] |
| Between 2,000 and 1,999,999 | 59.63% [59.06–60.19] | 56.58% [52.38–60.78] |
| Paris area | 15.02% [14.6–15.43] | 19.61% [16.13–23.09] |
| *Principal residence in priority neighborhood* | | |
| No | 95.69% [95.39–95.99] | 95.78% [94.06–97.51] |
| Yes | 4.31% [4.01–4.61] | 4.22% [2.49–5.94] |

*Pooled weighted % [95% confidence interval]*

*Individuals in each cell in each of the imputed data sets are available in* S3 Table

|  | Suicidal ideation in 2021 | |
| --- | --- | --- |
|  | **No (% [95% CI])** | Yes (% [95% CI]) |
| *Perceived health status* | | |
| Well to very well | 77.89% [77.35–78.42] | 72.81% [68.87–76.74] |
| Quite well | 18.65% [18.15–19.15] | 22.04% [18.33–25.76] |
| Poor to very poor | 3.46% [3.2–3.73] | 5.15% [3.21–7.09] |
| *Body mass index (kg/m$^2$)* | | |
| Under 18.5 | 3.32% [3.12–3.52] | 6.78% [4.56–8.99] |
| From 18.5 to under 25 | 50.82% [50.23–51.4] | 51.74% [47.54–55.94] |
| From 25 to under 30 | 30.94% [30.39–31.49] | 25.46% [21.79–29.13] |
| From 30 and over | 14.92% [14.47–15.37] | 16.03% [12.75–19.3] |

*(Continued)*

**Table 1.** (Continued)

|  | Suicidal ideation in 2021 | |
|---|---|---|
|  | **No (% [95% CI])** | Yes (% [95% CI]) |
| *Pre-pandemic somatic condition* |  |  |
| None | 60.92% [60.34–61.5] | 60.3% [56.25–64.34] |
| At least one | 39.08% [38.5–39.66] | 39.7% [35.66–43.75] |
| *Pre-pandemic history of mental health disorder* |  |  |
| None | 89.4% [89.02–89.79] | 65.27% [61.31–69.23] |
| At least one | 10.6% [10.21–10.98] | 34.73% [30.77–38.69] |
| *Tobacco use* |  |  |
| Never | 48.91% [48.33–49.49] | 41.55% [37.41–45.7] |
| Past only | 32.27% [31.74–32.8] | 33.47% [29.49–37.44] |
| Current | 18.82% [18.35–19.29] | 24.98% [21.27–28.69] |
| *Alcohol use* |  |  |
| Never | 29.38% [28.81–29.94] | 29.13% [25.23–33.02] |
| Rare | 13.57% [13.17–13.97] | 14.53% [11.42–17.64] |
| Occasional | 23.41% [22.93–23.88] | 20.31% [17.01–23.62] |
| Often | 23.1% [22.65–23.56] | 24.2% [20.7–27.7] |
| Daily | 10.54% [10.19–10.89] | 11.83% [9.22–14.44] |
| *Hospitalization rates in first LD department* |  |  |
| Lowest | 23.67% [22.86–24.47] | 23.06% [19.3–26.83] |
| Medium-low | 27.7% [26.49–28.91] | 26.84% [22.35–31.34] |
| Medium-High | 22.93% [22.02–23.83] | 23.2% [19.18–27.22] |
| Highest | 25.71% [25.04–26.38] | 26.9% [22.86–30.94] |

*Pooled weighted % [95% confidence interval]*

*Individuals in each cell in each of the imputed data sets are available in S3 Table*

*First LD: first lockdown (17/03/2020–11/05/2020)*

each covariate, as well as each modality of each covariate were below 10%, and (2) variance ratios of propensity scores computed after weighting were between 0.5 and 2 [15,20]. Covariates distribution after weighting was also assessed with chi-squared and Student *t* tests. Lastly, IPWeighted modified Poisson regression models with robust error variance were used to assess the association between both COVID-19 exposures in 2020 and suicidal ideation in 2021 [24]. If after IPWeighting residual distribution differences remained for some covariates, regression models were further adjusted for these incompletely balanced covariates. Models were therefore further adjusted for highest educational attainment for the self-reported COVID-19-like symptoms in 2020 model, and for highest educational attainment, perceived financial situation, household income, and residence in deprived neighborhood for the serology-confirmed SARS-CoV-2 infection in 2020 model.

**Sensitivity analyses.** To test the robustness of our results, we also estimated relative risks (RRs) by weighting the reference groups (no self-reported COVID-19-like symptoms; SARS-CoV-2-negative serology) to match the covariates' distribution in the exposed groups (respectively, self-reported COVID-19-like symptoms; serology-confirmed SARS-CoV-2 infection). This method, called average treatment effect on the treated (ATT; S5 Supporting Information), assesses what would have happened to participants who reported COVID-19-like symptoms, or who had a serology-confirmed SARS-CoV-2 infection, if they had not reported symptoms and been infected. The more consistent the estimated RRs in IPWeighted and ATTWeighted analyses, the more robust the results.

**Table 2.** Prevalence of serology-confirmed SARS-CoV-2 infection and covariates according to self-reported COVID-19-like symptoms in 2020, weighted by study weights, pooled from five imputed data sets.

| | Self-reported COVID-19 Symptoms in 2020 | |
| --- | --- | --- |
| | **No (% [95% CI])** | **Yes (% [95% CI])** |
| *Serology-confirmed SARS-CoV-2 infection in 2020* | | |
| No | 93.67% [93.36–93.97] | 69.19% [67.81–70.58] |
| Yes | 6.33% [6.03–6.64] | 30.81% [29.42–32.19] |
| *Self-reported COVID-19 symptoms in 2020* | | |
| No | | |
| Yes | | |
| *Gender* | | |
| Men | 46.79% [46.17–47.41] | 42.03% [40.51–43.55] |
| Women | 53.21% [52.59–53.83] | 57.97% [56.45–59.49] |
| *Age (years)* | | |
| 15–25 | 10.59% [10.21–10.96] | 15.62% [14.51–16.72] |
| 26–45 | 26.32% [25.8–26.85] | 37.72% [36.24–39.2] |
| 46–65 | 35.02% [34.46–35.58] | 34.24% [32.85–35.63] |
| >65 | 28.07% [27.45–28.69] | 12.42% [11.27–13.58] |
| *Place of birth* | | |
| Participant and parents born in mainland France | 82.51% [81.99–83.03] | 78.78% [77.37–80.19] |
| Participant or parents born in oversea territories | 1.99% [1.82–2.15] | 1.81% [1.43–2.19] |
| Participant born in France from parents born abroad | 8.3% [7.94–8.66] | 10.62% [9.6–11.64] |
| Participant born abroad | 7.2% [6.81–7.6] | 8.79% [7.69–9.88] |
| *Highest academic level* | | |
| No diploma | 8.41% [7.92–8.89] | 6.84% [5.72–7.95] |
| Certificate from junior high at most | 13.77% [13.25–14.29] | 10.26% [9.13–11.38] |
| Professional certificate | 19.72% [19.23–20.22] | 15.1% [13.97–16.22] |
| Baccalaureat | 19.31% [18.84–19.77] | 21.16% [19.95–22.36] |
| 2 to 4 years after bac | 24.77% [24.3–25.24] | 29.06% [27.78–30.34] |
| At least 5 years after bac | 14.02% [13.67–14.37] | 17.59% [16.6–18.58] |

Weighted % [95% confidence interval]

Individuals in each cell in each of the imputed data sets are available in *S4 Table*

First LD: first lockdown (17/03/2020–11/05/2020)

| | Self-reported COVID-19 Symptoms in 2020 | |
| --- | --- | --- |
| | **No (% [95% CI])** | **Yes (% [95% CI])** |
| *Employment status before first LD* | | |
| Employed | 46.75% [46.14–47.36] | 56.4% [54.86–57.94] |
| Students | 8.14% [7.83–8.45] | 12.78% [11.79–13.77] |
| Unemployed | 4.46% [4.17–4.75] | 5.75% [4.96–6.54] |
| Retired | 32.59% [31.98–33.21] | 15.7% [14.49–16.91] |
| Other situations | 8.06% [7.69–8.43] | 9.36% [8.33–10.39] |
| *Perceived financial situation* | | |
| Comfortable | 16.21% [15.81–16.6] | 15.75% [14.76–16.74] |
| Decent | 43.97% [43.36–44.57] | 39.51% [38.07–40.96] |
| Short | 31% [30.39–31.6] | 31.11% [29.66–32.56] |
| Hard or cannot make it without debt | 8.83% [8.41–9.25] | 13.63% [12.36–14.91] |
| *Household structure* | | |
| Single | 16.91% [16.38–17.45] | 15.51% [14.22–16.79] |

*(Continued)*

**Table 2.** (Continued)

| | Self-reported COVID-19 Symptoms in 2020 | |
|---|---|---|
| | **No (% [95% CI])** | **Yes (% [95% CI])** |
| Couple without children | 33.47% [32.89–34.05] | 23.27% [22.01–24.53] |
| Couple with children | 27.98% [27.46–28.5] | 33.76% [32.38–35.15] |
| Single-parent family | 6.95% [6.62–7.28] | 8.87% [7.99–9.75] |
| Child living at family house | 7.5% [7.18–7.82] | 10.19% [9.27–11.11] |
| Complex household | 7.18% [6.85–7.52] | 8.4% [7.58–9.22] |
| *Household income per consumption units* | | |
| Less resourceful | 13.48% [12.98–13.98] | 16.13% [14.8–17.46] |
| Medium-low | 16.48% [15.93–17.03] | 17.55% [16.2–18.9] |
| Medium | 20% [19.48–20.51] | 19.02% [17.75–20.3] |
| Medium-high | 23.77% [23.26–24.28] | 23.26% [21.98–24.53] |
| Wealthiest | 26.27% [25.8–26.75] | 24.04% [22.92–25.16] |
| *Physical space in usual accommodation* | | |
| No | 93.89% [93.56–94.22] | 89.63% [88.59–90.67] |
| Yes | 6.11% [5.78–6.44] | 10.37% [9.33–11.41] |
| *Changed accommodation for first LD* | | |
| No | 95.28% [95.02–95.54] | 93.34% [92.62–94.07] |
| Yes | 4.72% [4.46–4.98] | 6.66% [5.93–7.38] |
| *Access to safe outdoor space* | | |
| No | 8.37% [7.99–8.75] | 10.63% [9.6–11.66] |
| Yes | 90.69% [90.29–91.08] | 88.56% [87.49–89.62] |
| Other situations | 0.94% [0.8–1.08] | 0.82% [0.51–1.12] |
| *Usual area of residence* | | |
| Less affected area | 66.82% [66.24–67.4] | 58.1% [56.6–59.6] |
| Grand-Est | 8.58% [8.25–8.91] | 9.52% [8.71–10.32] |
| Hauts-de-France | 8.27% [7.92–8.63] | 8.8% [7.95–9.65] |
| Ile-de-France | 16.33% [15.88–16.78] | 23.59% [22.27–24.91] |
| *Urban density (urban units)* | | |
| oversea territories | 1.3% [1.18–1.42] | 1.04% [0.79–1.28] |
| Rural (less than 2,000) | 24.56% [24.04–25.09] | 20.82% [19.63–22] |
| Between 2,000 and 1,999,999 | 59.87% [59.27–60.47] | 57.64% [56.14–59.14] |
| Paris area | 14.27% [13.84–14.7] | 20.51% [19.26–21.76] |
| *Principal residence in priority neighborhood* | | |
| No | 95.82% [95.5–96.13] | 94.85% [93.93–95.77] |
| Yes | 4.18% [3.87–4.5] | 5.15% [4.23–6.07] |

Pooled weighted % [95% confidence interval]

Individuals in each cell in each of the imputed data sets are available in S4 Table

First LD: first lockdown (17/03/2020–11/05/2020)

| | Self-reported COVID-19 Symptoms in 2020 | |
|---|---|---|
| | **No (% [95% CI])** | **Yes (% [95% CI])** |
| *Perceived health status* | | |
| Well to very well | 78.1% [77.52–78.67] | 75.87% [74.45–77.3] |
| Quite well | 18.66% [18.12–19.2] | 19.03% [17.77–20.29] |
| Poor to very poor | 3.25% [2.98–3.51] | 5.1% [4.19–6] |
| *Body mass index (kg/m$^2$)* | | |
| Under 18.5 | 3.24% [3.03–3.46] | 4.29% [3.67–4.9] |

*(Continued)*

**Table 2.** (Continued)

| | Self-reported COVID-19 Symptoms in 2020 | |
|---|---|---|
| | No (% [95% CI]) | Yes (% [95% CI]) |
| From 18.5 to under 25 | 50.46% [49.84–51.09] | 53.24% [51.7–54.78] |
| From 25 to under 30 | 31.19% [30.61–31.78] | 28.6% [27.2–30] |
| From 30 and over | 15.1% [14.62–15.59] | 13.88% [12.77–14.98] |
| *Pre-pandemic somatic condition* | | |
| None | 60.63% [60.01–61.25] | 62.75% [61.25–64.25] |
| At least one | 39.37% [38.75–39.99] | 37.25% [35.75–38.75] |
| *Pre-pandemic history of mental health disorder* | | |
| None | 89.67% [89.27–90.07] | 84.57% [83.38–85.77] |
| At least one | 10.33% [9.93–10.73] | 15.43% [14.23–16.62] |
| *Tobacco use* | | |
| Never | 49.12% [48.5–49.74] | 46.6% [45.08–48.13] |
| Past only | 32.35% [31.78–32.91] | 31.92% [30.52–33.33] |
| Current | 18.53% [18.03–19.03] | 21.47% [20.18–22.76] |
| *Alcohol use* | | |
| Never | 29.22% [28.61–29.82] | 30.41% [28.93–31.89] |
| Rare | 13.59% [13.16–14.02] | 13.61% [12.55–14.66] |
| Occasional | 23.27% [22.77–23.77] | 23.9% [22.61–25.19] |
| Often | 23.05% [22.56–23.53] | 23.6% [22.4–24.79] |
| Daily | 10.88% [10.5–11.26] | 8.48% [7.66–9.31] |
| *Hospitalization rates in first LD department* | | |
| Lowest | 24.48% [23.66–25.3] | 18.25% [16.91–19.58] |
| Medium-low | 27.96% [26.76–29.16] | 25.87% [23.95–27.8] |
| Medium-High | 23% [22.12–23.87] | 22.51% [20.83–24.19] |
| Highest | 24.56% [23.9–25.22] | 33.37% [31.73–35.01] |

*Pooled weighted % [95% confidence interval]*

*Individuals in each cell in each of the imputed data sets are available in S4 Table*

*First LD: first lockdown (17/03/2020–11/05/2020)*

Interactions between gender, or age, and the two COVID-19 exposures in 2020 with respect to later suicidal ideation in 2021 were assessed but did not reached statistical significance (all *p*-value above 0.20).

In response to peer review comments, the following four sets of sensitivity analyses were also performed.

As pre-pandemic mental health disorder were partly assessed at the second follow-up, a memory bias cannot be ruled out. A sensitivity analysis removing the variable from propensity scores calculation was therefore performed.

As occurrence of COVID-19-like symptoms since December 2020, ascertained at the second follow-up in Summer 2021, can be associated with suicidal ideation between December 2020 and Summer 2021, a sensitivity analysis (i.e., propensity score calculation and IPWeighted Poisson regression model) was performed, removing participants who reported COVID-19-like symptoms between December 2020 and Summer 2021 (*n* = 4,564).

Time between self-reported COVID-19-like symptoms and suicidal ideation was also explored. We therefore conducted three sensitivity analyses ascertaining the association of self-reported COVID-19-like symptoms in 2020 with suicidal ideation in 2021, where COVID-19 symptoms were defined as follows: (1) self-reported COVID-19-like symptoms

**Table 3. Prevalence of self-reported COVID-19-like symptoms and covariates according to serology-confirmed SARS-CoV-2 infection in 2020, weighted by study weights, pooled from five imputed data sets.**

| | Serology-confirmed SARS-CoV-2 infection in 2020 | |
| --- | --- | --- |
| | No (% [95% CI]) | Yes (% [95% CI]) |
| *Self-reported COVID-19 symptoms in 2020* | | |
| No | 89.87% [89.52–90.23] | 57.41% [55.63–59.19] |
| Yes | 10.13% [9.77–10.48] | 42.59% [40.81–44.37] |
| *Gender* | | |
| Men | 46.34% [45.73–46.95] | 44.44% [42.62–46.26] |
| Women | 53.66% [53.05–54.27] | 55.56% [53.74–57.38] |
| *Age (years)* | | |
| 15–25 | 10.77% [10.41–11.14] | 15.78% [14.44–17.12] |
| 26–45 | 27.19% [26.67–27.71] | 33.87% [32.15–35.6] |
| 46–65 | 35.16% [34.61–35.71] | 32.61% [30.99–34.23] |
| >65 | 26.87% [26.27–27.47] | 17.74% [16.18–19.29] |
| *Place of birth* | | |
| Participant and parents born in mainland France | 82.81% [82.3–83.31] | 74.52% [72.76–76.29] |
| Participant or parents born in oversea territories | 1.89% [1.72–2.05] | 2.7% [2.14–3.25] |
| Participant born in France from parents born abroad | 8.22% [7.87–8.57] | 12.25% [10.96–13.54] |
| Participant born abroad | 7.08% [6.71–7.46] | 10.53% [9.12–11.94] |
| *Highest academic level* | | |
| No diploma | 8.23% [7.76–8.7] | 7.94% [6.51–9.38] |
| Certificate from junior high at most | 13.56% [13.06–14.06] | 10.92% [9.58–12.26] |
| Professional certificate | 19.56% [19.08–20.04] | 14.89% [13.64–16.14] |
| Baccalaureat | 19.38% [18.92–19.83] | 21.19% [19.71–22.66] |
| 2 to 4 years after bac | 25.05% [24.59–25.51] | 28.03% [26.52–29.54] |
| At least 5 years after bac | 14.23% [13.88–14.57] | 17.03% [15.88–18.18] |

Weighted % [95% confidence interval]

Individuals in each cell in each of the imputed data sets are available in S5 Table

First LD: first lockdown (17/03/2020–11/05/2020)

| | Serology-confirmed SARS-CoV-2 infection in 2020 | |
| --- | --- | --- |
| | No (% [95% CI]) | Yes (% [95% CI]) |
| *Employment status before first LD* | | |
| Employed | 47.45% [46.85–48.05] | 53.45% [51.61–55.28] |
| Students | 8.31% [8–8.62] | 12.96% [11.76–14.17] |
| Unemployed | 4.65% [4.37–4.94] | 4.39% [3.56–5.23] |
| Retired | 31.4% [30.8–32] | 20.49% [18.91–22.07] |
| Other situations | 8.18% [7.82–8.55] | 8.71% [7.56–9.86] |
| *Perceived financial situation* | | |
| Comfortable | 16% [15.61–16.38] | 17.59% [16.36–18.81] |
| Decent | 43.45% [42.86–44.04] | 42.68% [40.92–44.44] |
| Short | 31.27% [30.67–31.86] | 28.6% [26.88–30.32] |
| Hard or cannot make it without debt | 9.29% [8.87–9.71] | 11.13% [9.7–12.56] |
| *Household structure* | | |
| Single | 17.18% [16.65–17.71] | 12.44% [11.14–13.75] |
| Couple without children | 32.85% [32.29–33.41] | 25.23% [23.68–26.78] |
| Couple with children | 28.2% [27.69–28.7] | 33.96% [32.27–35.65] |
| Single-parent family | 7.12% [6.8–7.45] | 7.96% [6.96–8.96] |

*(Continued)*

**Table 3.** (Continued)

| | Serology-confirmed SARS-CoV-2 infection in 2020 | |
| --- | --- | --- |
| | **No (% [95% CI])** | **Yes (% [95% CI])** |
| Child living at family house | 7.53% [7.22–7.85] | 10.9% [9.78–12.03] |
| Complex household | 7.12% [6.8–7.43] | 9.5% [8.36–10.65] |
| *Household income per consumption units* | | |
| Less resourceful | 13.67% [13.19–14.16] | 15.3% [13.71–16.89] |
| Medium-low | 16.73% [16.19–17.27] | 15.64% [14.04–17.23] |
| Medium | 19.95% [19.45–20.45] | 19.08% [17.63–20.53] |
| Medium-high | 23.69% [23.2–24.19] | 23.77% [22.26–25.28] |
| Wealthiest | 25.95% [25.49–26.42] | 26.21% [24.8–27.62] |
| *Physical space in usual accommodation* | | |
| No | 93.82% [93.5–94.13] | 88.73% [87.33–90.13] |
| Yes | 6.18% [5.87–6.5] | 11.27% [9.87–12.67] |
| *Changed accommodation for first LD* | | |
| No | 95.21% [94.96–95.46] | 93.29% [92.38–94.2] |
| Yes | 4.79% [4.54–5.04] | 6.71% [5.8–7.62] |
| *Access to safe outdoor space* | | |
| No | 8.51% [8.14–8.88] | 10.19% [8.96–11.43] |
| Yes | 90.58% [90.19–90.97] | 88.76% [87.45–90.07] |
| Other situations | 0.91% [0.78–1.04] | 1.05% [0.54–1.56] |
| *Usual area of residence* | | |
| Less affected area | 66.85% [66.29–67.42] | 54.46% [52.64–56.28] |
| Grand-Est | 8.55% [8.23–8.87] | 10.12% [9.14–11.1] |
| Hauts-de-France | 8.22% [7.88–8.56] | 9.47% [8.35–10.6] |
| Ile-de-France | 16.37% [15.93–16.81] | 25.95% [24.29–27.6] |
| *Urban density (urban units)* | | |
| Oversea territories | 1.23% [1.12–1.35] | 1.53% [1.15–1.91] |
| Rural (less than 2,000) | 24.54% [24.03–25.06] | 19.58% [18.23–20.93] |
| Between 2,000 and 1,999,999 | 59.99% [59.4–60.58] | 55.65% [53.85–57.45] |
| Paris area | 14.23% [13.81–14.65] | 23.24% [21.64–24.83] |
| *Principal residence in priority neighborhood* | | |
| No | 95.96% [95.66–96.27] | 93.13% [91.92–94.34] |
| Yes | 4.04% [3.73–4.34] | 6.87% [5.66–8.08] |

*Weighted % [95% confidence interval]*

*Individuals in each cell in each of the imputed data sets are available in* S5 Table

*First LD: first lockdown (17/03/2020–11/05/2020)*

| | Serology-confirmed SARS-CoV-2 infection in 2020 | |
| --- | --- | --- |
| | **No (% [95% CI])** | **Yes (% [95% CI])** |
| *Perceived health status* | | |
| Well to very well | 77.28% [76.72–77.85] | 82.71% [81.2–84.22] |
| Quite well | 19.11% [18.58–19.64] | 14.89% [13.46–16.32] |
| Poor to very poor | 3.61% [3.33–3.89] | 2.4% [1.77–3.03] |
| *Body mass index (kg/m$^2$)* | | |
| Under 18.5 | 3.34% [3.13–3.55] | 3.76% [3.08–4.43] |
| From 18.5 to under 25 | 50.66% [50.05–51.27] | 52.41% [50.58–54.25] |
| From 25 to under 30 | 31.03% [30.46–31.6] | 29.13% [27.43–30.84] |
| From 30 and over | 14.97% [14.5–15.43] | 14.7% [13.29–16.1] |
| *Pre-pandemic somatic condition* | | |

*(Continued)*

**Table 3.** (Continued)

| | Serology-confirmed SARS-CoV-2 infection in 2020 | |
| --- | --- | --- |
| | No (% [95% CI]) | Yes (% [95% CI]) |
| None | 60.27% [59.66–60.87] | 66.95% [65.2–68.69] |
| At least one | 39.73% [39.13–40.34] | 33.05% [31.31–34.8] |
| *Pre-pandemic history of mental health disorder* | | |
| None | 88.83% [88.42–89.24] | 90.58% [89.46–91.7] |
| At least one | 11.17% [10.76–11.58] | 9.42% [8.3–10.54] |
| *Tobacco use* | | |
| Never | 47.86% [47.26–48.47] | 57.53% [55.75–59.32] |
| Past only | 32.47% [31.92–33.03] | 30.57% [28.95–32.19] |
| Current | 19.66% [19.16–20.16] | 11.9% [10.71–13.08] |
| *Alcohol use* | | |
| Never | 29.1% [28.51–29.69] | 31.95% [30.13–33.77] |
| Rare | 13.42% [13–13.84] | 15.18% [13.84–16.51] |
| Occasional | 23.44% [22.95–23.94] | 22.52% [21.1–23.94] |
| Often | 23.21% [22.74–23.69] | 22.24% [20.85–23.64] |
| Daily | 10.82% [10.45–11.19] | 8.11% [7.15–9.07] |
| *Hospitalization rates in first LD department* | | |
| Lowest | 24.45% [23.65–25.24] | 16.21% [14.6–17.82] |
| Medium-low | 28.11% [26.9–29.32] | 23.68% [21.59–25.77] |
| Medium-High | 22.77% [21.89–23.64] | 24.5% [22.49–26.5] |
| Highest | 24.68% [24.03–25.33] | 35.61% [33.63–37.59] |

*Weighted % [95% confidence interval]*

*Individuals in each cell in each of the imputed data sets are available in S5 Table*

*First LD: first lockdown (17/03/2020–11/05/2020)*

with last onset in the two months before first follow-up in Autumn 2020; (2) self-reported COVID-19-like symptoms with last onset four to six months before first follow-up in Autumn 2020; (3) self-reported COVID-19-like symptoms with last onset more than six months before first follow-up in Autumn 2020. In all three sensitivity analyses, the reference group reported no COVID-19-like symptoms.

Lastly, to explore the intersection of both COVID-19 exposures with respect to suicidal ideation, association of self-reported COVID-19-like symptoms before December 2020 with subsequent suicidal ideation in 2021 was separately assessed according to the SARS-CoV-2 serological status.

All the analyses were performed using SAS V9.4. Tests were two-sided and considered statistically significant at $p < 0.05$. Missing data on covariates (up to 4.84%) were handled using the fully conditional specification method and assuming that data were missing at random (SAS MI procedure, FCS statement, five imputed data sets; S4 Supporting Information). This study is reported as per the Strengthening the Reporting of Observational Studies in Epidemiology (STROBE) guideline (S1 Strobe Checklist).

## Results

### Main analysis

Among the 52,050 participants, 1.7% [1.5 to 1.8] ($n = 863$) reported suicidal ideation between December 2020 and Summer 2021, 9.6% [9.2 to 9.9] ($n = 5,091$) had a serology-confirmed SARS-CoV-2 infection in 2020 and 13.2% [12.9 to 13.6] ($n = 7,043$) reported COVID-19

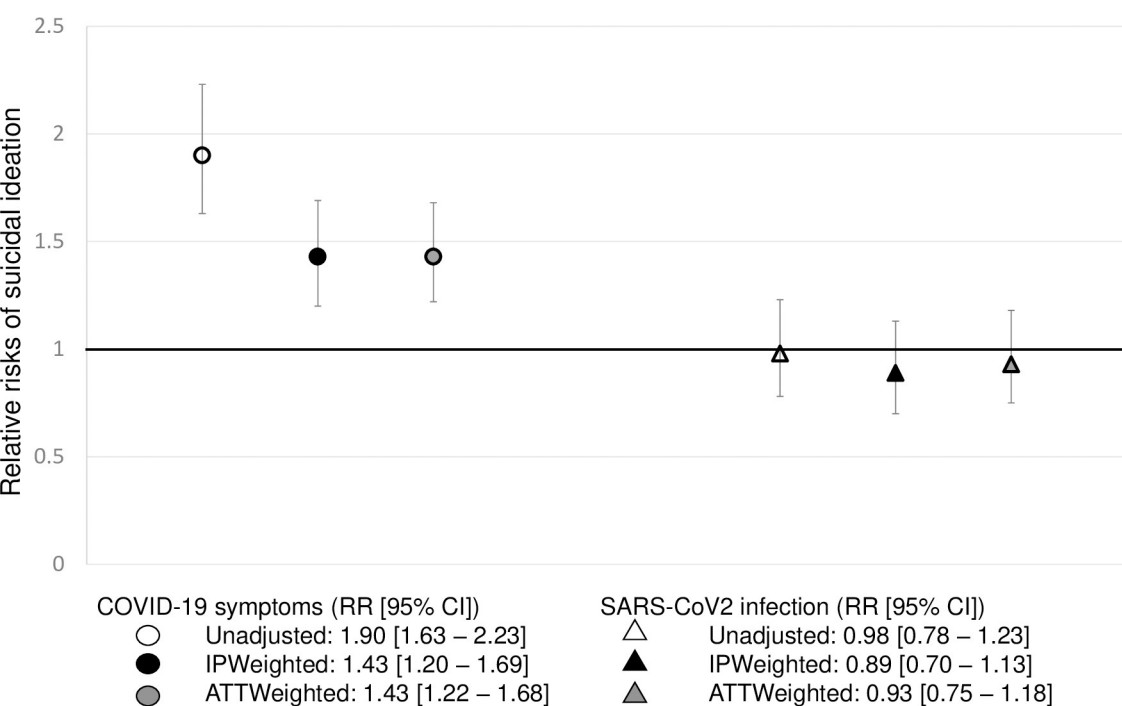

**Fig 3. Relative risk of suicidal ideation following self-reported COVID-19-like symptoms or SARS-CoV-2 infection in the EpiCov cohort.**

symptoms between February and November 2020. Of note, among participants who reported COVID-19-like symptoms, 30.8% [29.4; 32.2] had a serology-confirmed SARS-CoV-2 infection (Table 2). Among participants with a serology-confirmed SARS-CoV-2 infection, 42.6% [40.8; 44.4] reported COVID-19-like symptoms (Table 3). Using unadjusted modified Poisson regression model, self-reported COVID-19 symptoms were associated with a higher risk of suicidal ideation (RR [95% CI]: 1.90 [1.63 to 2.23]), while the same association was not observed for serology-confirmed SARS-CoV-2 infection (0.98 [0.78 to 1.23]).

As shown in Fig 3, after IPWeighting, self-reported COVID-19-like symptoms in 2020 were associated with a higher risk of suicidal ideation in 2021 (RR$_{ipw}$: 1.43 [1.20 to 1.69]), while serology-confirmed SARS-CoV-2 infection were not (RR$_{ipw}$: 0.89 [0.70 to 1.13]).

## Sensitivity analyses

Sensitivity ATTWeighted analyses yielded similar results as the IPWeighted ones. Self-reported COVID-like symptoms in 2020 were associated with a higher risk of suicidal ideation in 2021 (RR$_{watt}$: 1.43 [1.22 to 1.68]), while serology-confirmed SARS-CoV-2 infection in 2020 was not associated with suicidal ideation in 2021 (RR$_{watt}$: 0.93 [0.75 to 1.18]) (Fig 3).

When removing the pre-pandemic mental health disorder variable from propensity scores calculation, self-reported COVID-19-like symptoms in 2020 were associated with subsequent suicidal ideation in 2021 (RR$_{ipw}$ [95% CI]: 1.51 [1.27 to 1.79]), while serology-confirmed SARS-CoV-2 infections were not (RR$_{ipw}$: 0.87 [0.69 to 1.11]).

In line with main analyses, among participants who reported no COVID-19-like symptoms between December 2020 and Summer 2021, self-reported COVID-19-like symptoms in 2020 were associated with a higher risk of suicidal ideation in 2021 (RR$_{ipw}$: 1.40 [1.14 to 1.71]) (S6 Supporting Information).

Regarding the sensitivity analyses accounting for the period of the last symptom onset, among participants who reported COVID-19-like symptoms in 2020, 27.2% [25.9 to 28.6] reported a last onset in the two months before first follow-up in Autumn 2020, 25.8% [24.5 to 27.1] reported a last onset four to six months before first follow-up in Autumn 2020, and 47.0% [45.4 to 48.5] reported a last onset more than six months before first follow-up in Autumn 2020. A total of 19 participants had missing data on timing of the last COVID-19 symptoms onset. In line with main analyses, self-reported COVID-19-like symptoms with last onset in the two months before first follow-up were associated with a higher risk of later suicidal ideation in 2021 ($RR_{ipw}$ [95% CI]: 1.82 [1.39 to 2.39]). However, when the last onset of self-reported COVID-19-like symptoms was four to six months or more than six months before first follow-up, no association was found with suicidal ideation in 2021 ($RR_{ipw}$: 1.24 [0.91 to 1.67] and 1.24 [0.95 to 1.62], respectively) (S6 Supporting Information).

Lastly, the sensitivity analysis exploring the association of self-reported COVID-19-like symptoms with later suicidal ideation according to SARS-CoV-2 serology yielded similar results, although statistical significance was reached in one group but not the other ($RR_{ipw}$: 1.49 [0.95 to 2.32]), in the serology-confirmed SARS-CoV-2 infection group; $RR_{ipw}$: 1.49 [1.22 to 1.83], in the negative SARS-CoV-2 serology group) (S6 Supporting Information).

## Discussion

### Summary of findings

In a large French cohort from the general population, self-reported COVID-19-like symptoms in 2020, as defined by self-reports of sudden loss of taste/smell or fever alongside cough or shortness of breath or chest oppression, were associated with a higher risk of later suicidal ideation in 2021. Associations persisted after adjusting for a wide range of sociodemographic and health-related factors, using inverse probability weighting. Sensitivity analyses showed that participants with more recent symptoms were the most at-risk ones. When using serology-confirmed SARS-CoV-2 infection in 2020, as confirmed by circulating antibodies in dried-blood-sport samples collected for the cohort, no association was found with suicidal ideation in 2021. To the best of our knowledge, it is the first time the associations of self-reported COVID-19-like symptoms and serology-confirmed SARS-CoV-2 infection in 2020 with suicidal ideation in 2021 are assessed in a large, randomly selected, longitudinal study, using propensity score weighting methods.

### Serology-confirmed SARS-CoV-2 infection and suicidal ideation

Our findings suggest that SARS-CoV-2 is not likely to be involved in suicidal ideation. Although misclassification of infected individuals with very low levels of circulating antibodies cannot be ruled out, misclassification due to a new variant impairing effective detection of antibodies in blood samples seems unlikely as the first known variant of concern, the Alpha variant, was first observed in France at the very end of 2020 [25].

A recent study using two propensity score-matched cohorts from health databases found that the higher risk of mood and anxiety disorders seen after a SARS-CoV-2 infection was transient, with no evidence of a greater risk about 100 days (three months) after the infection, as compared with other respiratory infection [26]. The SARS-CoV-2 infection definition used was the International Classification of Disease 10th edition (ICD-10) code 07.1, i.e., COVID-19 confirmed by laboratory testing irrespective of severity of clinical signs or symptoms. Similar results were also found in the meta-analysis of five European cohorts were a higher risk of depressive symptoms was found in the first two months following a SARS-CoV-2 infection diagnosis (self-reported positive polymerase chain reaction (PCR) or antibody test), but not

after [12]. In the present study, time between SARS-CoV-2 infection and suicidal ideation could not be assessed. Nonetheless, the two COVID-19 waves in 2020 in France occurred in early Spring 2020 and Autumn 2020, while consultation in emergency department for suicide attempt in the first half of 2021 were on the rise in March and June. It is therefore possible that, in our study, the mean time between the SARS-CoV-2 infection detected by serology testing and suicidal ideation in 2021 exceeded two months, possibly explaining the absence of association found. Although we acknowledge that depressive symptoms do not always lead to suicidal ideation and that the link between suicidal ideation and suicidal attempt is not linear, our results seem in line with current findings.

## Self-reported COVID-19-like symptoms and suicidal ideation

Several hypotheses can explain the association between self-reported COVID-19-like symptoms and subsequent suicidal ideation. First are the consequences of a symptomatic illness on the quality of life. Symptomatic COVID-19 may require hospitalization or may have long-lasting symptoms, which both can affect mental health, whether directly or indirectly by, for example, affecting employment or financial outcome first [27–29]. A possible way of preventing symptomatic individuals to experience suicidal ideation could therefore be through policies aiming at mitigating the socioeconomical impact of the pandemic [30]. Second, anxiety and depressive symptoms could act as mediators of the relationship between COVID-19 and suicidal ideation [10,12,29]. Third, Paul and Fancourt showed that COVID-19 illness or death among friends/family or closed ones was associated with a higher risk of self-harm thoughts and behaviors [10]. This was also true for worries about relatives in the preceding week. As COVID-19 is a communicable disease, individuals with symptomatic COVID-19 could be more likely to have symptomatic cases among their relatives, increasing their risk of suicidal ideation, especially if they feel responsible for their relatives' infection.

In our sensitivity analyses, we found that participants with more recent onset of COVID-19-like symptoms in 2020 were at higher risk of suicidal ideation in 2021. But the association was not found in participants with older onset. In lines with other findings [12,26], the first few months after the infection seem to be a critical window for the onset of mental health disorders. The difference seen in statistical significance for the association of COVID-19-like symptoms with later suicidal ideation according to SARS-CoV-2 serological status is likely to be due a lack of statistical power to assess such association in the serology-confirmed SARS-CoV-2 infection group.

## Strengths and limitations

A main limitation of our study is the lack of prospectively collected pre-pandemic information as the EpiCov study was initiated in 2020. Nonetheless, many pre-pandemic characteristics were collected retrospectively and could be considered. Specific attention was given to reports of previous mental health disorders, as they are key predictors of suicidal behaviors [13]. We considered participants' history of anxiety, depression, or mental impairment, history of suicide attempt before November 2019, and self-report of a physician diagnosis of psychiatric disorders. Although less accurate than prospectively collected or health record data, these information give valuable insights of pre-pandemic mental health conditions.

A second limitation of our study is the use of a single question to assess suicidal ideation, which did not specifically explore whether suicidal ideation was active or passive, or whether the participants had an explicit suicidal plan. Scales specifically designed to assess suicidal ideation and behaviors [31–33] would have given a more accurate picture of the severity, chronicity, and intentionality of these thoughts and their association with COVID-19-like symptoms.

Regarding propensity scores, balance between persons who did or did not experience COVID-19 is only achieved for covariates included in propensity score estimations. Definition of relevant factors to assess COVID-19 disease and suicidal ideation was based on the existing scientific literature and availability of information in study questionnaires. The probability of imperfect balance due to unmeasured factors cannot be ruled out.

Lastly, although the EpiCov cohort is based on a randomly selected sample, attrition over-time as well as successive steps of participant selection for analyses purpose limit the generalizability of our results. Difference between included and excluded participants can be found in S1 Table. Of note, included participants were less likely to report suicidal ideation in 2021 but more likely to report COVID-19-like symptoms, potentially leading to an underestimation of the association between the two. Such difference was, however, not found for serology-confirmed SARS-CoV-2 infection.

## Conclusions

Self-reported COVID-19-like symptoms in 2020, especially recent ones, but not serology-confirmed SARS-CoV-2 infection in 2020 were associated with a higher risk of subsequent suicidal ideation in 2021 while adjusting for a wide range of sociodemographic and health-related factors using inverse probability weighting. In a pandemic context, bringing awareness on mental health in the first few months following a symptom onset could limit an increase in suicidal ideation. Mental health resources could, for example, be made available in COVID-19-related settings such as family doctors waiting room, pharmacies, or COVID-19 screening places. Examples of mental health resources include help lines numbers and websites, or brief and accessible description of what a deteriorated mental health can look like, and guidance on what to do when experiencing it.

## Supporting information

**S1 STROBE checklist. STROBE checklist for cohort studies.**
(DOCX)

**S1 Acknowledgements. Membership of the EpiCoV study group.**
(DOCX)

**S1 Supporting information. Detailed description of the pre-pandemic mental health disorders variable construction.**
(DOCX)

**S2 Supporting information. Evolution of total number of deaths by French departments during the first COVID-19 epidemic wave.**
(DOCX)

**S3 Supporting information. Directed acyclic graph.**
(DOCX)

**S4 Supporting information. Multiple imputation information.**
(DOCX)

**S5 Supporting information. Propensity score methodology.**
(DOCX)

**S6 Supporting information. Graphic representation of sensitivity analyses results.**
(DOCX)

**S1 Table. Comparison of EpiCov participants included versus excluded from analyses using two-sided chi-squared tests.**
(XLSX)

**S2 Table. Numerators and denominators of S1 Table.**
(XLSX)

**S3 Table. Numerators and denominators of the exposures and covariates, according to the onset of suicidal ideation in 2021, in each of the five imputed data sets.**
(XLSX)

**S4 Table. Numerators and denominators of covariates according to the COVID-19 symptoms variable in each imputed data set.**
(XLSX)

**S5 Table. Numerators and denominators of covariates according to the SARS-CoV-2 variable in each imputed data set.**
(XLSX)

**S6 Table. Covariates name and definition for multiple imputation.**
(XLSX)

**S7 Table. Parameters of the propensity scores balance for COVID-19-like symptoms.**
(XLSX)

**S8 Table. Parameters of the propensity scores balance for COVID-19-like symptoms for each covariate.**
(XLSX)

**S9 Table. Parameters of the propensity scores balance for SARS-CoV-2 serology.**
(XLSX)

**S10 Table. Parameters of the propensity scores balance for SARS-CoV-2 serology for each covariate.**
(XLSX)

## Author Contributions

**Conceptualization:** Camille Davisse-Paturet, Massimiliano Orri, Aline-Marie Florence, Jean-Baptiste Hazo, Marie-Claude Geoffroy, Maria Melchior, Alexandra Rouquette.

**Data curation:** Camille Davisse-Paturet, Aline-Marie Florence, Jean-Baptiste Hazo.

**Formal analysis:** Camille Davisse-Paturet.

**Funding acquisition:** Massimiliano Orri, Marie-Claude Geoffroy, Alexandra Rouquette.

**Investigation:** Jean-Baptiste Hazo, Josiane Warszawski.

**Methodology:** Camille Davisse-Paturet, Stéphane Legleye, Bruno Falissard, Maria Melchior, Alexandra Rouquette.

**Project administration:** Camille Davisse-Paturet, Alexandra Rouquette.

**Supervision:** Massimiliano Orri, Stéphane Legleye, Bruno Falissard, Marie-Claude Geoffroy, Maria Melchior, Alexandra Rouquette.

**Validation:** Camille Davisse-Paturet, Massimiliano Orri, Stéphane Legleye, Aline-Marie Florence, Jean-Baptiste Hazo, Josiane Warszawski, Bruno Falissard, Marie-Claude Geoffroy, Maria Melchior, Alexandra Rouquette.

**Visualization:** Camille Davisse-Paturet.

**Writing – original draft:** Camille Davisse-Paturet.

**Writing – review & editing:** Massimiliano Orri, Stéphane Legleye, Aline-Marie Florence, Jean-Baptiste Hazo, Josiane Warszawski, Bruno Falissard, Marie-Claude Geoffroy, Maria Melchior, Alexandra Rouquette.

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
