## [Editor Report · Decision Letter 0]

1 Jul 2022

Dear Dr Davisse-Paturet, 

Thank you for submitting your manuscript entitled "COVID-19 illness, SARS-CoV2 infection, and subsequent suicidal ideation in the French nationwide population-based EpiCov cohort: a propensity score analysis of more than 50,000 individuals." for consideration by PLOS Medicine.

Your manuscript has now been evaluated by the PLOS Medicine editorial staff as well as by an academic editor with relevant expertise and I am writing to let you know that we would like to send your submission out for external peer review.

Please re-submit your manuscript within two working days, i.e. by Jul 05 2022 11:59PM.

Kind regards,

Callam Davidson

Associate Editor

PLOS Medicine

---

## [Decision Letter · Decision Letter 1]

24 Aug 2022

Dear Dr. Davisse-Paturet,

Thank you very much for submitting your manuscript "COVID-19 illness, SARS-CoV2 infection, and subsequent suicidal ideation in the French nationwide population-based EpiCov cohort: a propensity score analysis of more than 50,000 individuals." (PMEDICINE-D-22-02146R1) for consideration at PLOS Medicine for our upcoming Special Issue. 

Your paper was evaluated by an associate editor and discussed among all the guest editors. It was also discussed with an academic editor with relevant expertise, and sent to independent reviewers, including a statistical reviewer. The reviews are appended at the bottom of this email and any accompanying reviewer attachments can be seen via the link below:

[LINK]

In light of these reviews, I am afraid that we will not be able to accept the manuscript for publication in the journal in its current form, but we would like to consider a revised version that addresses the reviewers' and editors' comments. Obviously we cannot make any decision about publication until we have seen the revised manuscript and your response, and we plan to seek re-review by one or more of the reviewers. 

We hope to receive your revised manuscript by Sep 14 2022 11:59PM. Please email us (plosmedicine@plos.org) if you have any questions or concerns.

We look forward to receiving your revised manuscript. 

Sincerely,

Callam Davidson, 

PLOS Medicine

plosmedicine.org

Comments from the Academic Editor

1) First I have some logical difficulty in understanding the examined cause-effect relationship. The outcome is the suicidal ideation between Dec 2020 and June 2021, based on a single question “Have you thought about committing suicide in the past year” (and then excluding those who said they did so in Nov 2020). The exposure is self-reported COVID-19 symptoms or serology-confirmed SARS-COV2 infection at the 2nd wave, hence between the beginning of the pandemic and Nov 2020.

Can one expect to have an increase (or decrease) in thinking about committing suicide in, say, May 2021 when they had COVID-19 symptoms or SARS-COV2 infection in May 2020? If any such cause-effect is present, would it not be more natural to expect such in June-November 2021 but how are such possibilities accounted for or not accounted for in the present analyses?

In a similar vein, if someone had symptoms or infection in Jan 2021, it could have had an effect on suicidal ideation between Jan and June 2021 (this is a hypothesis that the study is examining), but where are such possibilities accounted for in the present analyses?

2) As suggested by Reviewer #3, I think the timeline of the study is essential to understanding all these and an expanded version of their Supplementary Fig 1 must be included in the main text.

3) Like Reviewer #2, I also have a difficulty calling this study as nationally representative, when only 16% of the originally nationally representative population is analyzed.

4) Looking at their Supplementary Fig 1, the variables (pre-pandemic diagnoses measured at the 3rd wave) are included in calculating the IPW. Sure, they are supposed to be pre-baseline but they are subjectively recalled at the same time as the outcome measurement. I wonder if the authors need not report at least a sensitivity analysis excluding this variable in the IPW calculation.

5) Although the statistical reviewer is basically OK with the analyses, I am not sure if the authors have done their best to impute the missing data. When the authors can use multiple imputation, is it justified to exclude almost 30000 (so amounting to some 60% of the final sample) for missing that variable (while they must have had other data because these 30000 did respond in principle to the surveys). Likewise some 2400 people are excluded because they had missing data on suicidality (but apparently had other data): in this situation, could they also not impute the missing outcome, using some auxiliary variables?

6) I agree with Reviewer #3 that the relationship (overlaps) between those with self-reported COVID-19 symptoms and those with serologically confirmed SARS-COV2 infection. Could they be non-overlapping? If so, what could each mean, especially the former? Could some further analysis focusing on those with self-reported symptoms but without serological confirmation shed some more light?

Requests from the editor

Throughout: Consider whether the term ‘COVID-19 illness’ ought to be updated to ‘COVID-19 symptoms’ to describe more accurately what was assessed in the study.

Title: Related to the above, please revise your title to “COVID-19 symptoms, SARS-CoV2 infection, and suicidal ideation in France: A population-based propensity score–matched cohort study”, or similar.

Please structure your abstract using the PLOS Medicine headings (Background, Methods and Findings, Conclusions).

Abstract Methods and Findings:

* Please include the length of follow up.

* In the last sentence of the Abstract Methods and Findings section, please describe the main limitations of the study's methodology.

The Data Availability Statement (DAS) requires revision. If the data are owned by a third party but freely available upon request, please note this and state the owner of the data set and contact information for data requests (web or email address). Note that a study author cannot be the contact person for the data.

Citations should be preceding punctuation.

Please include paragraphs in your Introduction. 

The terms gender and sex are not interchangeable (as discussed in https://www.who.int/health-topics/gender); please use the appropriate term.

Ethics: Please specify whether informed consent was written or oral.

Please remove the Role of the Funding Source section. 

Please remove the Authors contribution and Data Sharing, and Acknowledgements section from the main text as all the relevant information is captured as metadata via the Submission Form. You are welcome to still include an Acknowledgements section but it should not contain details of study funders.

Please remove italic formatting in the References and include the first six author names before et al.

Please do not report P<0.0001, report instead as P<0.001 (Supporting Information). 

Please ensure that the study is reported according to the STROBE guideline, and include the completed STROBE checklist as Supporting Information. Please add the following statement, or similar, to the Methods: "This study is reported as per the Strengthening the Reporting of Observational Studies in Epidemiology (STROBE) guideline (S1 Checklist)."

Did your study have a prospective protocol or analysis plan? Please state this (either way) early in the Methods section.

Comments from the reviewers:

Reviewer #1: Alex McConnachie, Statistical Review

Davisse-Paturet et al present an impressive analysis of data from a national cohort study, looking at the association between COVID-19 and subsequent suicidal ideation. This review considers the use of statistics in the paper.

The basic method of deriving models to predict the propensity of having COVID-19 illness or SARS-Cov-2 infection, and using these to weight the analysis of each exposure variable and the outcome, is good. I thought the paper was clearly written, and the results are well presented and interpreted, so my comments are fairly minor.

One exception is the use of the E-value. Whilst I think it is a nice idea, the more I read about it, the less convinced I am that it really adds very much. Ultimately, it is simply a transformation of the estimated association between exposure and outcome, so I think it could be excluded without affecting the paper.

Lines 184-188 describe the methods for dealing with missing covariate data. Personally, I think if you are going to that much trouble, it is worth doing multiple imputation of the missing data, so as to allow for the additional uncertainty due to the missingness.

Lines 213-214 say that the estimated association between COVID-19 illness and suicidal ideation was unchanged after IPW. I would disagree. The estimate reduces from 1.90 to 1.43, so it had roughly halved after reweighting. In Figure 2, there is a visible reduction in the estimated association. The authors describe the adjusted association of 1.43 as "an almost 1.5 higher risk", which I think is a slight exaggeration. I would just say "a 43% higher risk".

Did the authors consider a combined model, adjusting for both COVID-19 illness and SARS-Cov-2 infection? Was there a big overlap between the two? Could there be an interaction?

Line 234: These sorts of analysis have come to be known as "causal" analyses, but I think this is a misleading term. These methods are a good way to adjust an analysis of the association between an exposure and an outcome, but the final estimated association cannot be described as "causal". As with any observational study, the possibility of unobserved confounders will always remain.

In Figure 1, the final set of exclusions are described as having "Insufficient 'Do not wish to answer' to be analysed". Is the word "Insufficient" correct here?

I would rather see Table 1 with separate columns for those with and without the outcome, and for those who are exposed/unexposed. The "by-outcome" table is given in table S2, but could "by-exposure" tables also be shown?

Finally, and I hope this is just a typo, but in the supplement, the weights for the exposed group are given as 1/(1-p), and for the unexposed as 1/p. I believe it should be the other way round. Those who were exposed, despite having a low probability of being exposed (i.e. small p) should be up-weighted in the analysis, as should those who were unexposed, despite a high probability of being exposed (high p, small 1-p).

Also, I prefer "exposed/unexposed" over "case/control" in this context, since this is not a case-control study.

Reviewer #2: I read the manuscript "COVID-19 illness, SARS-CoV2 infection, and subsequent suicidal ideation in the French nationwide population-based EpiCov cohort: a propensity score analysis of more than 50,000 individuals" with great interest and would like to share the following few thoughts with the authors. 

1. The authors found that self-reported COVID-19 symptoms are indicative of suicidal ideation whereas serology-confirmed SARS-CoV2 infection was not. Before we speculate the underlying reasons for such disparity, I wonder if the authors could clarify the number of participants with both self-reported symptoms and serology-confirmed infection. Namely, among the 9.57% with a SARS-CoV2 infection in 2020 and 13.23% that reported COVID-19 symptoms, how many had both? 

2. Related to this, can the authors provide a comparison between the above two groups, something similar as the Supplementary table 2? This will likely help us to understand the differences between these two groups of participants.

3. According to Supplementary Table 1, the participants included in the final analysis are rather different from the participants excluded from the analysis. Specifically, there seems to be a higher prevalence of self-reported COVID-19 symptoms, but a lower prevalence of suicidal ideation, among the included participants, compared with excluded participations. This will likely lead to an underestimated association between self-reported symptoms and suicidal ideation. This is however not the case for serology-determined infection.

4. Related to the above, I think the authors cannot claim their data as population-based, because the study participants included in the analyses are not representative of the underlying source population. This should rather be discussed as a limitation of the study.

Reviewer #3: This manuscript examines the impact of COVID-19 illness and SARS-CoV2 infection on suicidal ideation in a cohort, EpiCOV, which is a longitudinal nationwide French cohort aimed at examining the impact of the pandemic on daily life and health of individuals. The study consisted of three waves of data collection. The sample size analyzed was 52,050 participants aged 15 years and older who had participated in the second wave of follow-up and participated in dried-blood-spot sampling kit to test for antibodies against SARS-COV2's spike protein S1 domain and answered questions regarding suicidal ideation. The survey asked the following question at the 3rd wave of the study: "Since December 2020, have you thought about committing suicide (yes/no)?". There were also questions about thoughts of suicide in the last 12 months and lifetime suicide attempts. Self-reported symptoms of COVID-19 were assessed and defined as a self-report of any unusual episode of sudden loss of taste/smell or any unusual episode of fever alongside a cough, shortness of breath, or chest oppression. The study is timely and addresses an important research question that has not been studied yet. The large sample size, longitudinal design, examining serology to test for COVID-19 antibodies, and the use of propensity scores to control for covariates and potential confounders are all major strengths. Below are some concerns that if addressed, it would strengthen the study. 

- The manuscript is not clear on the temporality of assessments between exposure and outcome. It would help to include a figure to describe the waves of the study and when questions were asked about exposure and outcome. The outcome is described at multiple points in the manuscript with questions about suicidal ideation since the pandemic and in the last 12 months. 

- The authors do not describe the baseline characteristics of the sample analyzed vis-a-vis the rest of the sample who were not included in the analyses. It is difficult to determine potential ascertainment biases without this comparison. 

- It is not clear how the authors assessed prior history of suicidal behavior and mental disorders. It would help to include more details and whether there were screens for depression and anxiety symptoms. 

- It is not clear why the authors excluded those with prior history of suicidal behavior between November 2019 and 2020 when someone could reattempt and those with a prior history are at highest risk for re-attempt. 

- It would be important to include more details about the study weights and how they were computed. 

- The description of the sample's characteristics is based on some subgroups when there's variability (e.g., non-smokers when there were a good proportion of the sample who were past or current smokers). 

- One of the limitations of the study is the assessment of ideation and attempt using individual items. The questions do not differentiate between passive and active suicidal ideation; and whether the suicide attempts were actual vs. gestures and/or preparatory behaviors. The authors need to refer to suicide attempt as suicidal behavior as the latter is often referring to actual suicide attempt, which was not assessed here.

[LINK]

---

## [Decision Letter · Decision Letter 2]

24 Nov 2022

Dear Dr. Davisse-Paturet,

Thank you very much for re-submitting your manuscript "Suicidal ideation following self-reported COVID-19 like symptoms or serology-confirmed SARS-CoV2 infection in France: a propensity score weighted analysis from a cohort study" (PMEDICINE-D-22-02146R2) for review by PLOS Medicine.

I have discussed the paper with my colleagues and the academic editor and it was also seen again by two reviewers. I am pleased to say that provided the remaining editorial and production issues are dealt with we are planning to accept the paper for publication in the journal.

[LINK]

We look forward to receiving the revised manuscript by Dec 01 2022 11:59PM.   

Sincerely,

Callam Davidson, 

Associate Editor 

PLOS Medicine

plosmedicine.org

Requests from Editor:

Please address the comment from the statistical reviewer.

The Data Availability Statement (DAS) requires revision. For each data source used in your study: 

b) If the data are owned by a third party but freely available upon request, please note this and state the owner of the data set and contact information for data requests (web or email address). 

Please correct 'SARS-CoV2' to 'SARS-CoV-2' throughout.

Please structure your abstract using the PLOS Medicine headings (Background, Methods and Findings, Conclusions). Please combine the Methods and Findings sections into one section, “Methods and findings”.

Please update the beginning of your abstract to read: 'A higher risk of suicidal ideation associated with self-reported COVID-19 like symptoms or COVID-19 infection has been observed in cross sectional studies, but evidence from longitudinal studies remains limited. This study's aims...' 

In the last sentence of the Abstract Methods and Findings section, please describe the main limitations of the study's methodology (the sentence that currently finishes the abstract 'Methods' can be relocated).

In the Abstract Conclusions, please avoid vague statements such as those in the concluding sentence of the abstract. Instead, mention specific implications substantiated by the results.

Please include a basic demographic summary of your sample in the Methods and Findings of your Abstract.

Please structure your author summary using 2-3 single sentence bullet points for each of the three questions. Bullet points should be objective, brief, succinct, specific, accurate, and avoid technical language.

Line 46: 'dose-response fashion'.

The terms gender and sex are not interchangeable (as discussed in https://www.who.int/health-topics/gender); please consider whether sex (and thus male/female) would be the more appropriate term dependent on how this covariate data was collected.

Please cite your Supporting Information as outlined here: https://journals.plos.org/plosmedicine/s/supporting-information

For international readers unfamiliar with how residential areas in France were affected during the pandemic, it may be useful to provide further details regarding this categorisation (i.e., brief overview of Grand-Est, Hauts-de-France-Ile-de-France).

Line 226: 'sensitivity analysis'

Lines 229-241: This reads as a description of the sensitivity analysis results. Please only describe methodology in the Methods section and relocate description of findings to the Results section.

Please update your STROBE checklist to use section names and paragraph numbers, as page/line numbers are likely to change. 

Table 1 does not appear to be cited anywhere in the main text.

Please present numerators and denominators for percentages, at least in the Tables.

Please define the abbreviation DOM in Table legends.

The term 'mental health behaviour' may present a translation issue as the meaning is ambiguous. Please provide a definition of this term. 

Please update 'associated to' to 'associated with' throughout.

Please consider tabulating the results of your sensitivity analyses and presenting these in the Supporting Information.

Lines 359-361: I feel this should be in the Limitations section.

Your concluding sentence (line 395-396) introduces the concept of socioeconomic involvement, despite this not being a key finding of your study. I would suggest moving this sentence earlier in the Discussion.

Please carefully check your references for formatting issues, I noted problems with references 5, 12, and 22.

Reference 23 is a preprint and ought to be identified as such (see https://journals.plos.org/plosmedicine/s/submission-guidelines#loc-references for further guidance).

Comments from Reviewers:

Reviewer #1: Alex McConnachie, Statistical Review

I thank Davisse-Paturet et al for their responses to my original comments, and I am mostly happy with these.

The one exception is their handling of the association between COVID-19 illness and suicidal ideation separately in those with and without confirmed SARS-Cov-2 infection. The authors state that there was an association in the uninfected subgroup, but not the infected subgroup. However, the RRs are identical in the two groups. The fact that the p-value is significant for one but not the other is not the correct way to assess these associations. It would be better to fit a single model and test the interaction between illness and infection; given the results presented, this will be non-significant, meaning that there is no evidence to suggest that the association between COVID-19 illness and suicidal ideation is any different in those with or without confirmed infection.

Reviewer #2: Thank you for the responses to my comments.

[LINK]

---

## [Decision Letter · Decision Letter 3]

3 Jan 2023

Dear Dr. Davisse-Paturet,

Thank you very much for re-submitting your manuscript "Suicidal ideation following self-reported COVID-19 like symptoms or serology-confirmed SARS-CoV-2 infection in France: a propensity score weighted analysis from a cohort study" (PMEDICINE-D-22-02146R3) for review by PLOS Medicine.

Before we can accept the paper for publication in the journal, we require that you address a few remaining issues.

The remaining issues that need to be addressed are listed at the end of this email. Please take these into account before resubmitting your manuscript.

We look forward to receiving the revised manuscript by Jan 10 2023 11:59PM.   

Sincerely,

Callam Davidson, 

Associate Editor 

PLOS Medicine

plosmedicine.org

Requests from Editors:

Please update your abstract introduction such that the first sentence reads: 'A higher risk of suicidal ideation associated with self-reported COVID-19 like symptoms or COVID-19 infection has been observed in cross sectional studies, but evidence from longitudinal studies remains limited. The aims of this study were twofold...'.

The final sentence of the abstract introduction (‘Propensity scores…’) can be deleted as this information is already present in the abstract methods and findings.

The additional demographic data included in the abstract can be moved to the beginning of the abstract methods and findings, i.e., ‘52,050 participants from the French EpiCov cohort were included (median follow-up time = 13.7 months). In terms of demographics, 53.84% were women…’ etc. 

Please update your abstract conclusions such that the first sentences read: ‘Self-reported COVID-19-like symptoms in 2020, but not serology-confirmed SARS-CoV-2 infection in 2020, were associated with a higher risk of subsequent suicidal ideation in 2021. The exact role of SARS-CoV-2 infection with respect to suicide risk has yet to be clarified’.

In the interest of brevity, please delete the two final sentences in the Abstract (beginning ‘In France…’).

In the Author Summary, please expand your response to the second question (‘What Did the Researchers Do and Find?’) slightly to include brief details of the study design as well as headline numbers relating to the key findings. 

I think that the first bullet point under the question ‘Who do these findings mean’ should be rephrased to avoid misinterpretation. Please update to ‘Individuals experiencing COVID-19 like symptoms in the first year of the pandemic were at higher risk of later suicidal ideation, but this association was not observed for serologically confirmed SARS-CoV-2 infection, thus further study is needed to confirm the role of the virus in relation to suicide risk.’

Line 20: Please update ‘distress’ to ‘poor mental health’. 

The term ‘sensibility analysis’ appears several times throughout the text – please review and correct to ‘sensitivity analysis’ where appropriate. 

Comments from Reviewers:

Reviewer #1: Alex McConnachie, Statistical Review

Thank you once again for considering my comments. I had not considered the implications of fitting an interaction model whilst using IPTW. Whilst I am sure there are ways around this, I am happy with the changes that the authors have made, and have no further comments.

[LINK]

---

## [Editor Report · Decision Letter 4]

10 Jan 2023

Dear Dr Davisse-Paturet, 

On behalf of my colleagues and the Academic Editor, Dr Toshi Furukawa, I am pleased to inform you that we have agreed to publish your manuscript "Suicidal ideation following self-reported COVID-19 like symptoms or serology-confirmed SARS-CoV-2 infection in France: a propensity score weighted analysis from a cohort study" (PMEDICINE-D-22-02146R4) in PLOS Medicine.

When making the formatting changes, please also make the following editorial changes:

* Please format your Author Summary using bullet points (see previously published PLOS Medicine articles for an idea of the expected format).

* Please update 'Summer 2021' to 'July 2021' in your Author Summary.

* Please update the second bullet under question two of your Author Summary to read ‘Among these participants, reporting COVID-19 like symptoms in 2020 was associated with a higher risk of reporting suicidal ideation in 2021 (relative risk [95% confidence interval] 1.43 [1.20 – 1.69]) while having a serologically confirmed SARS-CoV-2 infection in 2020 was not associated with a higher risk of reporting suicidal ideation in 2021. These results account for sociodemographic and health-related factors.’

PRESS

We frequently collaborate with press offices. If your institution or institutions have a press office, please notify them about your upcoming paper at this point, to enable them to help maximise its impact. If the press office is planning to promote your findings, we would be grateful if they could coordinate with medicinepress@plos.org.

Thank you again for submitting to PLOS Medicine. We look forward to publishing your paper and will be in touch soon with further information regarding the planned publication window for the Special Issue. 

Sincerely, 

Callam Davidson 

Associate Editor 

PLOS Medicine